# Carbon- and System-Aware LoRA Scaling for On-Device LLMs via Hierarchical Multi-Objective Reinforcement Learning

## Abstract

On-Device deployment of large and small language models (LLMs / SLMs) faces critical challenges in balancing performance, energy consumption, and carbon footprint on various mobile and wearable devices. We introduce a hierarchical multi-objective reinforcement learning approach for dynamic Low-Rank Adaptation (LoRA) scaling that optimizes carbon efficiency as the primary objective while maintaining acceptable performance and energy consumption. Our method employs Proximal Policy Optimization (PPO) with a carbon-first reward function that prioritizes carbon efficiency (inferences per mg $CO_2$) and then energy efficiency (inferences per Joule). Across smartwatches, AR glasses, VR headsets and tablets running DistilGPT2, OPT-125M, DialoGPT-Small, and GPT-2, our approach achieves an average of 20.5 inf/J (smartwatch) and up to a peak of 35.1 inf/J in optimal configurations, as well as up to 0.412 perf/mg $CO_2$. These results demonstrate the effectiveness of carbon-aware optimization for sustainable edge AI.

## 1 Introduction

The proliferation of on-device large and small language model (LLM/SLM) applications has created an urgent need for deployment strategies that balance computational performance with environmental sustainability. Although prior work emphasizes energy efficiency or model compression Strubell et al. (2019); Schwartz et al. (2020), the carbon footprint of edge inference, including operational and embodied emissions, remains underexplored Henderson et al. (2020).

Low-Rank Adaptation (LoRA) Hu et al. (2021) enables efficient adaptation, but choosing *where* and *how* to adapt (which layers) across heterogeneous devices/tasks is nontrivial. We introduce hierarchical RL for dynamic LoRA scaling, where the agent adaptively selects the number of transformer layers equipped with related LoRA adapters and hyper-parameters, instead of using a fixed configuration.

## 2 Methodology

### 2.1 Problem Formulation

We pose dynamic LoRA scaling as multi-objective optimization: the agent selects a subset/number of LoRA layers $l \in [l_{\min}, l_{\max}]$ for device $p$, model $m$, and task $t$ to maximize

$$\pi^*(s) = \arg\max_{\pi} \mathbb{E}_{\pi}\big[R_{\text{hier}}(s, a)\big], \tag{1}$$

$$R_{\text{hier}} = w_c R_c + w_e R_e \cdot \mathbb{I}(R_c \geq \tau) + w_p R_p - \sum_i w_i P_i, \tag{2}$$

where $R_c, R_e, R_p$ are carbon, energy, and performance rewards; $\mathbb{I}(\cdot)$ enforces a carbon threshold $\tau$; $P_i$ penalize constraint violations of system metrics including temperature, latency, memory and power.

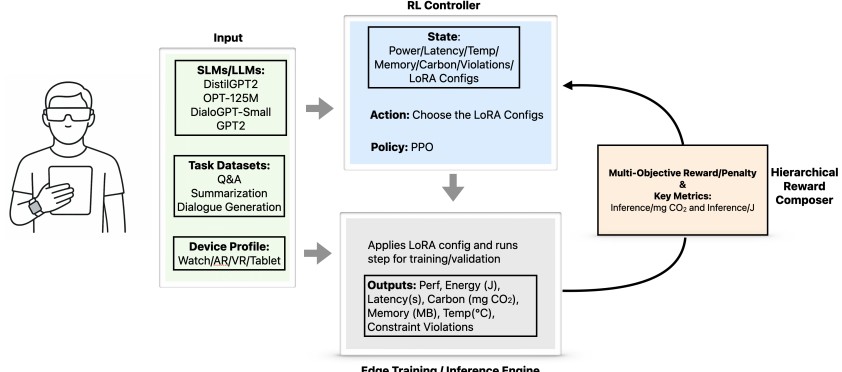

Figure 1: System overview. The PPO agent selects LoRA configs under carbon-first reward while the simulator provides device-aware feedback (power/thermal/memory/carbon).

## 2.2 CARBON-FIRST HIERARCHICAL REWARD

Primary (carbon): $\eta_c = \frac{1}{\text{mg CO}_2/\text{inf}} = \frac{\text{inf/s}}{\text{mg CO}_2}$ with

$$R_c = \frac{1}{1 + c_{\text{actual}}/c_{\text{target}}}, \quad c_{\text{target}} = 0.35 \text{ mg CO}_2. \tag{3}$$

Secondary (energy): $R_e = \tanh\left(\frac{\text{inf/J}}{12.0}\right) \cdot \alpha(R_c)$, where $\alpha(R_c) = 1$ if $R_c \geq 0.55$ else $0.25$. And tertiary (task): $R_p = \text{clip}(\text{summary\_score}, 0, 1)$.

## 2.3 EDGE HARDWARE SIMULATION

We model the carbon contribution of four device profiles (Watch/AR/VR/Tablet) with:

$$C_{\text{operational}} = E_{\text{inf}} \cdot I_{\text{grid}}, \quad C_{\text{manufacturing}} = \frac{C_{\text{device}} \cdot 1000}{N_{\text{lifetime}}}, \quad C_{\text{total}} = C_{\text{operational}} + C_{\text{manufacturing}} \tag{4}$$

where $I_{grid}$ is grid carbon intensity (mg/Wh), $C_{device}$ is device manufacturing carbon (g), and $N_{lifetime}$ is expected lifetime inferences. Other system metrics can be modeled based on the emprical formulas, for example, the temperature can be modelled as:

$$T_{\text{surface}} = T_{\text{amb}} + \alpha \cdot P_{\text{consumed}} \cdot R_{\text{thermal}}, \tag{5}$$

with safety limits: watch/AR $\leq 42°$C; VR $\leq 40°$C Henderson et al. (2020).

## 2.4 IMPLEMENTATION DETAILS: LORA CONFIGURATION AND DYNAMIC SELECTION

**Configuration.** We implement LoRA via PEFT Mangrulkar et al. (2022). Global hyperparameters that can be tuned are defined in `config.py` (e.g., rank $r = 8$, $\alpha = 16$, dropout $= 0.05$; device-specific $[l_{\min}, l_{\max}]$ ranges). In `edge_training.py`, we construct:

$$\text{LoraConfig}(r, \alpha, \text{dropout}, \text{target\_modules}=\{\text{q\_proj}, \text{v\_proj}\}),$$

and wrap the base LM with `get_peft_model` on the selected LoRA configs.

**Dynamic layer selection.** In `rl_environment.py`, the PPO agent outputs an action that maps to either (i) a count of layers to adapt (respecting the device's $[l_{\min}, l_{\max}]$), and/or (ii) a specific subset of layer indices. The environment applies LoRA on those layers, executes the task, and computes rewards (carbon→energy→performance). This closes the loop between policy, LoRA placement, and device-aware feedback.

**Algorithm 1** Hierarchical Carbon-First PPO Training

---

1: Initialize policy $\pi_\theta$, value $V_\phi$
2: Initialize environment (devices, models, tasks)
3: **for** $i = 1$ to $N$ **do**
4:     Collect trajectories $\tau = \{(s_t, a_t, r_t)\}$ with $\pi_\theta$
5:     Map $a_t$ to LoRA *layer count/subset*; apply PEFT to those layers
6:     Compute $R_c, R_e, R_p$ and $R_{\text{hier}}$; record constraint penalties
7:     Update policy/value with PPO Schulman et al. (2017) (we use SB3 Raffin et al. (2021))
8: **end for**
9: Evaluate across device–model–task grid

---

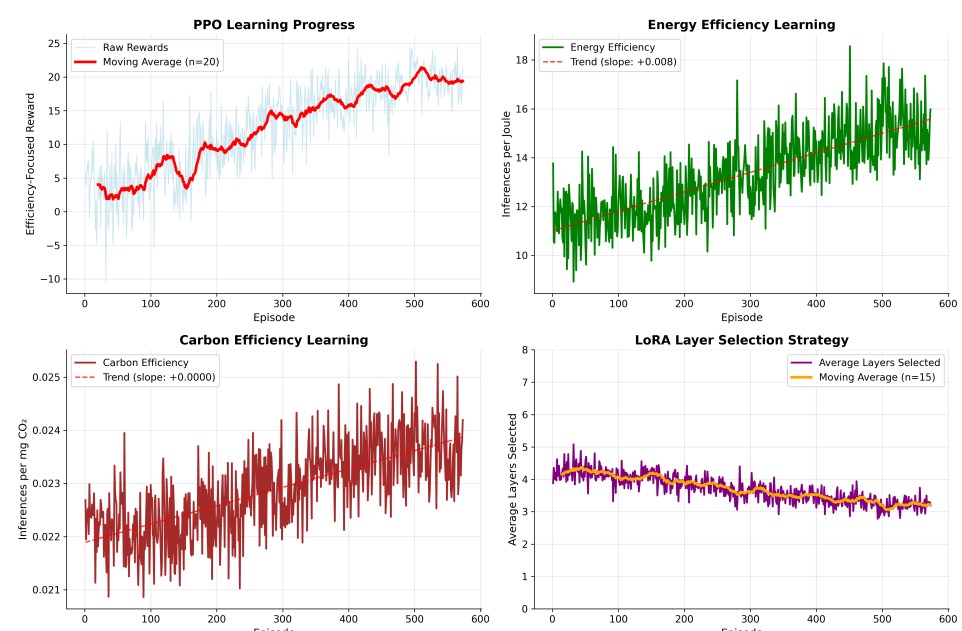

Figure 2: **PPO learning curves** for energy/carbon efficiency and learned LoRA-layer strategy. The agent converges to ∼3-4 adapted layers on average, balancing performance and carbon.

## 3 RESULTS

### 3.1 RL LEARNING AND CONVERGENCE

Figure 2 shows PPO learning curves: episode rewards increase from 39.1 to 43.3 (500 episodes), meanwhile, carbon and energy efficiency trend has both positive slope but with variance especially carbon slope is smaller, reflecting competing objectives, and more tuning on the reward function and training episodes are needed. The learned policy reaches up to 35.1 inf/J and 0.412 perf/mg $CO_2$ on tablet-like configurations.

### 3.2 DEVICE-SPECIFIC EFFICIENCY AND LAYER PATTERNS

Table 1 and Figure 3 summarize device-level outcomes. Watches peak in energy efficiency (20.5 inf/J) with 1 adapted layer; tablets peak in carbon efficiency (0.412 perf/mg $CO_2$) with 2 layers. Figure 4 details the learned *frontier* across devices.

## 4 DISCUSSION AND LIMITATIONS

This work initially study carbon-first hierarchical RL framework that learns where to place LoRA adapters on-device. Some limitations need to be resolved as follows. **Scope:** Evaluation spans 4

Table 1: Mean performance by device. Values are averages across models/tasks. Peak efficiencies (e.g., smartwatch = 35.1 inf/J) exceed these means and are reported separately in Appendix.

| Device | Energy Eff. (inf/J) | Carbon Eff. (perf/mg $CO_2$) | Optimal Layers |
|---|---|---|---|
| Smartwatch | 20.5 | 0.142 | 1 |
| AR Glasses | 9.9 | 0.340 | 2 |
| Tablet | 6.4 | 0.412 | 2 |
| VR Headset | 9.9 | 0.384 | 3 |

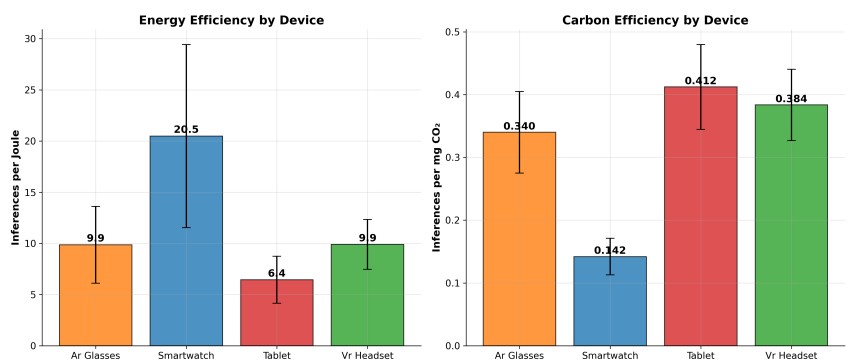

Figure 3: (a) Carbon and (b) energy efficiency by device. The learned policies favor few-layer adaptation (1–3) with device-specific optima, yielding strong carbon gains on tablets and energy gains on watches. Error bars: ± one standard deviation (std) of the distribution of metrics across all runs (models × tasks × configs) for that device.

models and 3 tasks; broader coverage and seed sweeps are future work. **Modeling:** Carbon/thermal models are simplified; real devices will refine intensities and transfer coefficients Henderson et al. (2020). **Validation:** Hardware-in-the-loop measurements are needed to verify LoRA–carbon causality. **Training:** Variance in learning suggests sensitivity to PPO and simulator settings; multi-objective RL or evolutionary strategies could further stabilize Van Moffaert & Nowé (2014); Parisi et al. (2014).

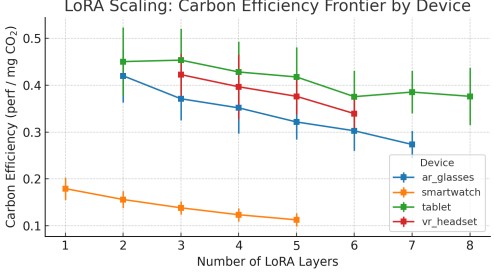
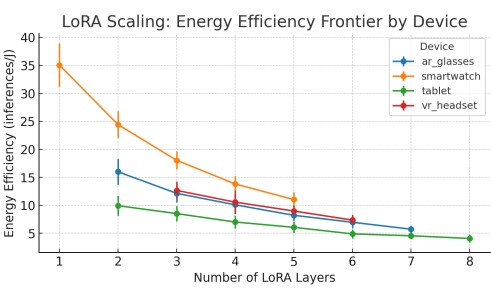

(a) Carbon efficiency frontier vs. adapted layers     (b) Energy efficiency frontier vs. adapted layers

Figure 4: Layer-scaling frontiers. Carbon/energy gains saturate beyond 3–4 layers, motivating *dynamic* (few-layer) selection instead of uniform deep adaptation.

## 5 CONCLUSION

We introduced a hierarchical reinforcement learning approach for dynamic LoRA scaling that prioritizes carbon and energy efficiency in on-device LLM deployment. Our method achieves significant environmental benefits while maintaining competitive energy efficiency (up to 35.1 inf/J) and high constraint satisfaction rates. The learned policies demonstrate intelligent adaptation to diverse device capabilities, providing a foundation for environmentally conscious edge AI systems.

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

# A  TECHNICAL APPENDICES

## A.1  EXPERIMENTAL SETUP

**Models:**

- DistilGPT2 (82M parameters)
- OPT-125M (125M parameters)
- DialoGPT-Small (117M parameters)
- GPT-2 (124M parameters)

**Tasks:**

- Question Answering (SQuAD dataset)
- Text Summarization (CNN/DailyMail dataset)
- Dialogue Generation

**Devices and LoRA Layer Ranges:**

- Smartwatch: 1–5 layers (limited by power/thermal constraints)
- AR Glasses: 2–7 layers (moderate computational capacity)
- VR Headset: 3–6 layers (balanced power/performance profile)

- Tablet: 2–8 layers (highest computational capacity)

The numbers indicate the range of transformer layers that can be equipped with LoRA adapters on each device type, constrained by device-specific power budgets, thermal limits, and memory capacity.

## A.2 TRAINING CONVERGENCE

Training Curves shows good convergence with representative configs: LoRA $r=8$, $\alpha=16$, dropout 0.05.

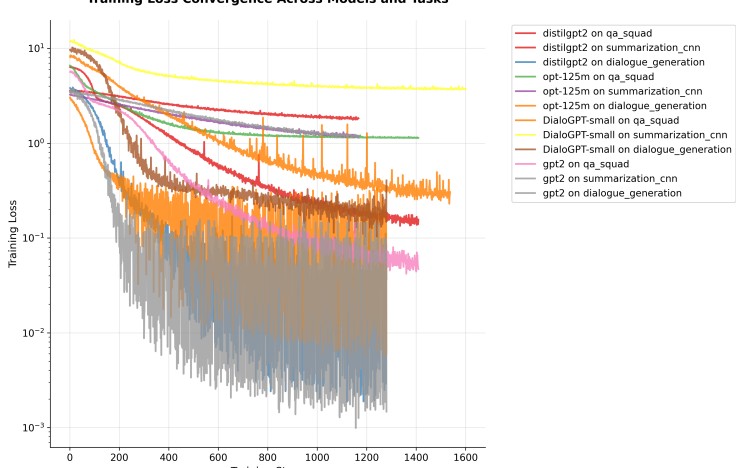

Figure 5: Representative LoRA training-loss traces across models/tasks.

## A.3 CARBON FOOTPRINT DISTRIBUTION

The carbon footprint analysis reveals significant variability across device-model-task configurations, as shown in Figure 6. The distribution exhibits a mean of 46.6 mg $CO_2$ per inference with substantial variance, indicating that optimal LoRA layer selection critically depends on the specific deployment context. The left panel shows the frequency distribution of carbon emissions, with most configurations clustering around 35-50 mg $CO_2$/inf. The right panel demonstrates the performance-carbon trade-off across devices, where the Pareto frontier clearly separates efficient configurations from suboptimal ones. Notably, tablets and VR headsets show wider carbon footprint ranges due to their higher computational capacity, while smartwatches cluster toward lower emissions but also lower performance scores.

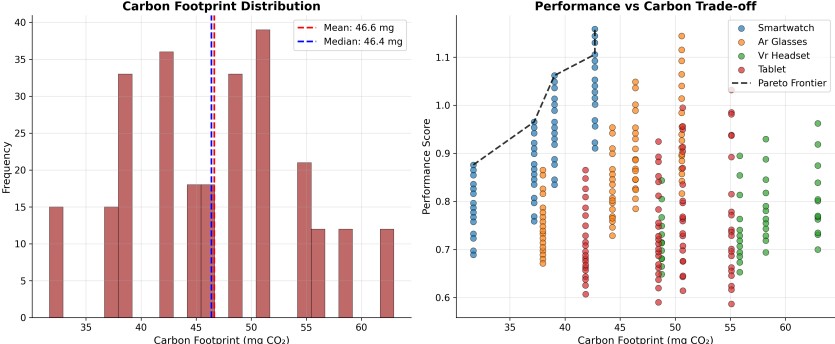

Figure 6: Carbon distribution (mean 46.6 mg $CO_2$/inf) with substantial cross-configuration variance.

## A.4 CONSTRAINT SATISFACTION

Device-specific constraint analysis demonstrates the effectiveness of our hierarchical reward structure in respecting hardware limitations, as illustrated in Figure 7. The left panel shows constraint satisfaction rates across four categories: memory, power, latency, and temperature. Simpler devices (smartwatch, AR glasses) achieve near-perfect constraint satisfaction due to their conservative LoRA layer limits and lower computational demands. However, more capable devices (tablets, VR headsets) experience constraint violations, particularly in temperature and power domains when the RL agent pushes toward higher performance configurations. The right panel quantifies total constraint violations, showing 24 violations for tablets and 33 for VR headsets, primarily occurring during aggressive few-layer LoRA adaptation that maximizes carbon efficiency at the cost of thermal stability. This validates our carbon-first reward design but highlights the need for stricter constraint penalties in future work.

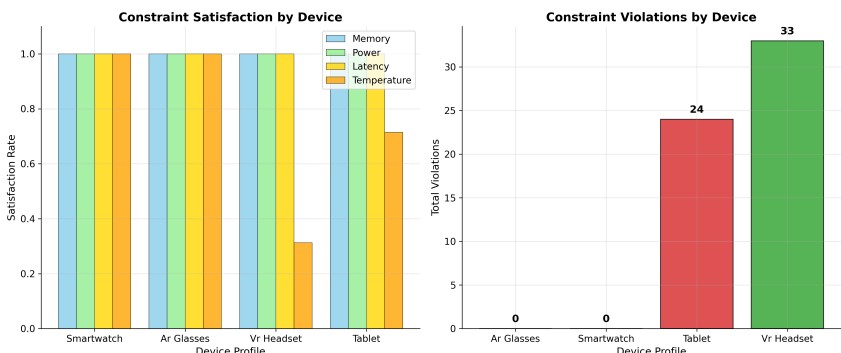

Figure 7: Constraint satisfaction: perfect for simpler devices; more violations on tablet/VR under heavy loads.

## A.5 CARBON-EFFICIENCY SCALING LAWS

Across profiles, optimal adapted-layer counts cluster at 1–2 (wearables) and 2–3 (mobile). Returns diminish beyond 4 layers due to overhead, supporting dynamic few-layer selection.

