# OpenReview forum: "Carbon- and System-Aware LoRA Scaling for On-Device LLMs via Hierarchical Multi-Objective Reinforcement Learning"
_ICLR.cc/2026/Conference — Submitted to ICLR 2026_

### Official Review · Reviewer_qgac · 2025-10-28

**Soundness:** 2
**Presentation:** 2
**Contribution:** 1
**Rating:** 2
**Confidence:** 4

**Summary:**

The paper presents a hierarchical multi-objective reinforcement learning approach for optimizing lora scaling in on-device large and small LLMs. The primary objective is to improve carbon efficiency while maintaining acceptable energy consumption and performance on mobile and wearable devices. The authors use PPO and a carbon-first reward function, achieving significant environmental benefits in devices.

**Strengths:**

1. The application of hierarchical reinforcement learning to optimize LoRA scaling for environmental efficiency is a novel approach in the context of edge AI.
2. The methodology is well-explained.
3. The focus on carbon efficiency is highly relevant in the context of growing concerns about the environmental impact of AI and on-device models.

**Weaknesses:**

1. The evaluation is limited to only a few devices and models.
2. The reliance on simplified carbon/thermal models weakens the claim of practical applicability.

**Questions:**

See above.

---

### Official Review · Reviewer_9Yrc · 2025-10-29

**Soundness:** 2
**Presentation:** 2
**Contribution:** 1
**Rating:** 2
**Confidence:** 3

**Summary:**

This paper proposes a hierarchical multi-objective reinforcement learning framework for dynamic LoRA (Low-Rank Adaptation) scaling on edge devices running small language models. The approach uses Proximal Policy Optimization (PPO) to learn which transformer layers should be equipped with LoRA adapters, optimizing primarily for carbon efficiency (inferences per mg CO₂), secondarily for energy efficiency (inferences per Joule), and tertiarily for task performance. The method is evaluated across four device types (smartwatch, AR glasses, VR headset, tablet) running four small models (DistilGPT2, OPT-125M, DialoGPT-Small, GPT-2) on three tasks (QA, summarization, dialogue). Results show the learned policies achieve up to 35.1 inf/J energy efficiency and 0.412 perf/mg CO₂ carbon efficiency, with optimal configurations using 1-3 adapted layers depending on the device

**Strengths:**

- Timely and important topic: Addresses carbon footprint of edge AI, which is increasingly critical as LLM deployment scales to billions of devices. I can understand the motivation and basic takeaways even just by reading through the abstract and into.

- Novel problem formulation: First work to frame dynamic LoRA layer selection as a carbon-first multi-objective RL problem with hierarchical rewards. Frankly speaking, this is a **rare angle**

- Comprehensive device modeling: Considers multiple device types with realistic constraints (thermal, power, memory, latency) and models both operational and embodied carbon

**Weaknesses:**

- Simulation-only evaluation: All results are from simulated environments without any hardware-in-the-loop validation, making carbon/energy claims unverified

- Oversimplified carbon modeling: Equations 4-5 use basic linear models that don't capture real device behavior, grid intensity variations, or manufacturing complexity

- Limited experimental scope: Only 4 models, 3 tasks, 4 device types tested; no statistical significance testing or confidence intervals reported (only std dev in Figure 3)

- Incomplete convergence: Figure 2 shows high variance, and the paper admits "more tuning on the reward function and training episodes are needed"

**Questions:**

- Reward function design: How were the weights (wc, we, wp) and threshold τ in Equation 2 chosen? What happens if you vary these?

- Carbon threshold: Why 0.55 for Rc threshold in the energy reward function? What's the sensitivity to this choice?

- Action space: How exactly does the agent select layers—as a count or specific indices? Section 2.4 mentions both but doesn't clarify which was used.

- Constraint violations: Figure 7 shows substantial violations for tablets/VR. Why not increase constraint penalty weights wi in Equation 2?

- Convergence: Figure 2 shows episode rewards plateauing around 20-22 after 600 episodes. Why continue to 600 if no improvement?

- Layer selection patterns: Does the agent learn meaningful patterns (e.g., always adapt attention layers) or is selection arbitrary? Can you visualize which layers are selected?

- Real-world deployment: How would this work in practice—retrain for each user? Use pre-trained policies? Adapt online?

- Comparison with model compression: How does dynamic LoRA compare with pruning, quantization, or knowledge distillation for carbon efficiency?

---

### Official Review · Reviewer_P4Jd · 2025-10-31

**Soundness:** 2
**Presentation:** 1
**Contribution:** 3
**Rating:** 2
**Confidence:** 5

**Summary:**

This paper introduces a hierarchical multi-objective reinforcement learning approach for dynamic for scaling LoRA models that optimizes carbon efficiency as the primary objective, employing PPO with a carbon-first reward function. In their approach, the agent adaptively selects the optimal number of layers and LoRA adapters and hyper-parameters

**Strengths:**

- It is a contribution towards measuring and optimizing the carbon footprint of edge inference, including operational and embodied emissions, which is underexplored

-The problem being addressed is an important one, given the ubiquity of edge inference

- The method has environmental benefits while maintaining  energy efficiency

**Weaknesses:**

- The paper doesn't provide enough detail about the methodology or results to judge their relevance

- The four device profiles are simulations, not actual tests

- There aren't enough tasks and empirical data to make meaningful conclusions

- More work needs to be done in order to make the paper more compelling and ready for publication.

**Questions:**

- Why weren't real devices tested, why did you do simulations?

- Is the number of layers simply dependent on the memory of the device?

-Why would someone deploy LoRA models on a smartwatch?

---

### Official Review · Reviewer_8apc · 2025-11-01

**Soundness:** 1
**Presentation:** 1
**Contribution:** 2
**Rating:** 2
**Confidence:** 4

**Summary:**

This paper proposes to use reinforcement learning (RL) for multi-objective optimization for choosing the layers for LoRA fine-tuning. The goal is to optimize carbon, energy, and performance together under a carbon threshold and system constraints such as temperature, latency, memory, and power. The paper describes the formulation and the high-level equations for each component, shows the training curves, and the results (# of layers, carbon efficiency, and energy efficiency).

**Strengths:**

The problem of how to design a model (in this case, LoRA) that optimizes multiple objectives under realistic system constraints is an important problem to study. The sustainability (carbon efficiency) is a relatively new problem that deserves more studies. The proposed approach to use RL for this optimization problem seems reasonable.

**Weaknesses:**

While the high-level goal of the problem seems good, the paper appears unfinished. There are many aspects of the work that need either further development or more detailed description/discussion.

1) Motivation/Problem Statement
While the high-level optimization objectives make sense, I don't think the paper sufficiently motivates the problem in the context of LoRA fine-tuning. For example, it does not appear that a LoRA configuration will affect manufacturing carbon emissions. The motiviation will be more compelling if the paper first shows how much LoRA configurations can effect carbon, energy, and performance.

2) Related Work
Multi-objective optimization in general and optimizing an ML model is a heavily studied topic. RL is also a popular approach for optimization. The paper needs to include a discusison on the related work and explicitly point out how this work is different/better than the existing state-of-the-art.

3) Details of the Proposed Scheme
The paper only describes the optimization formulation at a high-level. It is not clear how the individual components such as carbon emission, energy consumption, performance, etc. were obtained. It is also not clear which models and use cases were studied and which datasets were used. The paper needs to provide more technical details. Ideally, it will be also helpful to explicitly point out which parts of the proposed approach are new and novel.

4) Evaluation
The current evaluation simply reports the results obtained with the proposed scheme. In order to show that the proposed scheme improves the state-of-the-art. The evaluation needs to compared the results for the proposed method with other existing techniques.

**Questions:**

See the comments in the weakness section.

---

### Meta-Review · Area_Chair_VbyQ · 2025-12-25

**Summary:**

All reviewers agree that the paper addresses an important and timely problem, and that framing dynamic LoRA layer selection as a carbon-first multi-objective RL problem is interesting. However, there is strong consensus that the work is not yet mature. The main issues are:

* insufficient methodological detail and clarity
* simulation-only evaluation with oversimplified carbon/energy models,
* lack of comparisons to existing baselines (e.g. static LoRA, pruning, quantization)
* limited experimental scope and validation.

As a result, reviewers question both soundness and practical relevance, leading to rejection recommendations.

**Reviewer Concerns:**

None (no rebuttal submitted).

**Reviewer Scores:**

Since no rebuttal, I'd expect no change to the scores: 2 / 2 / 2 / 2

---

### Decision · Program_Chairs · 2026-01-26

Reject